

# Reconstructing unreadable QR codes: a deep learning based super resolution strategy

Yasin Sancar

Open Education Faculty, Atatürk University, Erzurum, Yakutiye, Turkey

## ABSTRACT

Quick-response (QR) codes have become an integral component of the digital transformation process, facilitating fast and secure information sharing across various sectors. However, factors such as low resolution, misalignment, panning and rotation, often caused by the limitations of scanning devices, can significantly impact their readability. These distortions prevent reliable extraction of embedded data, increase processing times and pose potential security risks. In this study, four super-resolution models Enhanced Deep Super Resolution (ESDR) network, Very Deep Super Resolution (VDSR) network, Efficient Sub-Pixel Convolutional Network (ESPCN) and Super Resolution Convolutional Neural Network (SRCNN) are used to mitigate resolution loss, rotation errors and misalignment issues. To simulate scanner-induced distortions, a dataset of 16,000 computer-generated QR codes with various filters was used. In addition, super-resolution models were applied to 4,593 QR codes that OpenCV's QRCodeDetector function could not decode in real-world scans. The results showed that EDSR, VDSR, ESPCN and SRCNN successfully read 4,261, 4,229, 4,255 and 4,042 of these QR codes, respectively. Furthermore, the EDSR, VDSR, ESPCN and SRCNN models trained by OpenCV's deep learning-based WeChat QR Code Detector function to read 2,899 QR codes that were initially unreadable and simulated on the computer were able to successfully read 2,891, 2,884, 2,433 and 2,560 of them, respectively. These findings show that super-resolution models can effectively improve the readability of degraded or low-resolution QR codes.

## INTRODUCTION

In an era where fast and reliable data transmission is becoming increasingly critical, QR codes have emerged as a widely utilized tool across various domains. Due to their high data storage capacity and ease of use, QR codes play a crucial role in sectors such as commerce, transportation, healthcare, and data security, serving essential functions including data tracking, authentication, and information sharing (*Tarjan et al., 2014*; *Victor, 2012*; *Hernandez et al., 2023*). However, the accuracy and readability of QR codes can be compromised by factors such as the resolution limitations of scanning devices, scanner-induced blurring, and physical wear and tear. Such degradations can significantly reduce the perceptibility of QR codes, leading to data loss and disruptions in operational

Corresponding author
Yasin Sancar,
yasinsancar@atauni.edu.tr

processes (*Tkachenko et al., 2015*; *Krombholz et al., 2015*; *Kato et al., 2011*). In particular, poor-quality scans or physically damaged QR codes may pose serious risks to the reliability of commercial and operational workflows (*Shi & Feng, 2024*).

To address these challenges, super-resolution technologies have emerged as a promising solution for enhancing the visual quality and readability of QR codes. Super-resolution techniques enable the reconstruction of lost details by upscaling low-resolution images to higher resolutions (*Xiao et al., 2023*; *Ran et al., 2023*; *Tian, 2014*). Recent advancements in image enhancement have demonstrated the significant potential of deep learning-based super-resolution models in restoring fine details and correcting distortions (*Shi et al., 2022*; *Ko et al., 2021*; *Ha et al., 2018*). Among these, models such as Enhanced Deep Super-Resolution (ESDR) network, Very Deep Super-Resolution (VSDR) network, Efficient Sub-Pixel Convolutional Network (ESPCN), and Super-Resolution Convolutional Neural Network (SRCNN) are particularly notable for their ability to reconstruct intricate details, especially in small and complex regions. For instance, EDSR minimizes resolution loss in QR codes through its deep residual network structure (*Zhang et al., 2022*; *Omar, Daniel & Rosbi, 2024*; *Esmaeilzehi, Ahmad & Swamy, 2020*; *Birdal, 2024*), while VDSR achieves high performance through its deep layer architecture and fast computational capabilities (*Wang et al., 2023*; *Jaiseeli & Raajan, 2024*).

ESPCN, another prominent model in the super-resolution domain, is particularly effective in applications requiring real-time performance, such as recovering small details in tiny patterns and sharp edges within QR codes critical for accurate decoding (*Ko et al., 2021*; *Jiang, Lin & Shang, 2021*; *He, 2009*). Similarly, SRCNN provides an efficient approach for improving QR code readability by successfully enhancing the details of low-resolution images despite its relatively simple architecture (*Sun, Pan & Tang, 2022*; *Velumani et al., 2022*).

QR code recognition is a fundamental aspect of automatic data extraction, and OpenCV provides two widely used tools for this purpose: QRCodeDetector and WeChatQRCodeDetector. The standard QRCodeDetector in OpenCV offers a lightweight and efficient method for detecting and decoding QR codes from images. However, its performance highly depends on image quality, often struggling with low-resolution, blurred, or distorted QR codes. In contrast, WeChatQRCodeDetector integrates deep learning-based image enhancement techniques, significantly improving robustness against real-world distortions such as blur, low contrast, and perspective transformations. Despite its superior performance, WeChatQRCodeDetector is not entirely error-free, and its effectiveness diminishes when QR codes are subjected to severe degradation (*Ma et al., 2023*).

In this study, a comparative analysis of super-resolution models for improving the readability of QR codes is performed. The main objective is to meet the demands for traceability, security, and data accessibility while facilitating more reliable data transmission in commercial and operational environments by improving QR code resolution. To this end, the performance of EDSR, VDSR, ESPCN, and SRCNN is evaluated to determine the most effective super-resolution approach to QR code enhancement. This analysis aims to identify the most suitable model to correct distortions

and improve readability, providing a valuable reference for future developments in this field.

### Research contributions

This research presents a comprehensive analysis that evaluates the performance of different super resolution models to improve the readability of QR codes in low resolution scans. The main contributions of our research can be summarized as follows:

- Comparative model analysis for low-resolution QR codes: A comparative evaluation of the success of common super-resolution models such as EDSR, VDSR, ESPCN, and SRCNN in improving the readability of QR codes. This analysis aims to determine the most effective method by examining in detail the capacity of each model to enhance QR code resolution.
- Investigating the relationship between resolution degradation and model performance: This study investigates the impact of different degradations in the resolution of QR codes on model performance. It reveals how factors such as blurring, geometric deformation, and resolution loss encountered during the scanning process change the impact of each super-resolution model on readability.
- Choosing the most effective model for image enhancement: The research goes further to guide on ways to ensure that there is reliable data transfer in commercial and operational processes through the identification of the most suitable super-resolution model to be used in QR code applications. Against this background, concrete recommendations have been developed about the use of super-resolution in QR code applications.
- Contribution for future applications: This research provides a basis for further development efforts to enhance the capacity of QR codes to meet reliable data transfer and traceability requirements. The comparative approach and model analyses used in the study demonstrate the potential of using super-resolution models for the enhancement of low-resolution QR codes and other image enhancement efforts.

## RELATED STUDIES

As QR codes become increasingly prevalent, recent studies have focused on their structure, recognition, and security aspects. Researchers have examined different approaches to improve QR code robustness, enhance readability, and explore novel applications in various domains (*Udvaros & Szabó, 2024*).

*Tsai, Lee & Chen (2023)* It explored the use of deep learning techniques to analyze QR codes and focused on source identification based on print characteristics. In their work, they investigated various convolutional neural network (CNN) architectures, including ResNet and MobileNet, to discriminate between QR codes printed by different devices. By leveraging advanced machine learning models, they showed that subtle variations in printed QR codes can be used to trace their origins, which has important implications for security, fraud detection and forensic analysis.

Image enhancement and super-resolution have become crucial research areas, particularly for improving the readability of low-resolution QR codes and minimizing data

loss. Various studies have explored classical and deep learning-based approaches to enhance QR code readability and overall image quality. For instance, *Tarjan et al. (2014)* investigated the factors influencing QR code readability after scanning, analyzing the sensitivity of QR codes to scanner quality, resolution, and distortion. Their findings emphasized the necessity of high-quality QR codes to ensure reliable data transmission in commercial traceability applications.

However, most existing studies focus on general image enhancement techniques rather than specialized solutions for degraded QR codes. While deep learning-based super-resolution methods have been extensively explored for photographic images, their effectiveness in improving QR code readability has not been systematically evaluated. This study addresses this research gap by conducting a comparative analysis of four state-of-the-art super-resolution models, EDSR, VDSR, ESPCN, and SRCNN, on QR codes affected by real-world distortions such as blur, noise, and contrast variations. By evaluating the recovery performance of these models in terms of scannability, this study provides insights into the most effective approaches for improving QR code readability in practical applications.

*Jin et al. (2023)* proposed a fast and blind sharpening algorithm based on local maximum and minimum intensity prioritization to remove blur from QR code images. Based on the principle of decreasing local maxima and increasing minima in the blurring process, the method quickly analyzes these intensity values through binarization. The use of thresholding instead of classical quadratic decomposition as an alternative decomposition approach provided advantages in terms of both accuracy and processing time. In experiments on real fuzzy QR codes, this method produced sharper results than conventional maximum *a posteriori* (MAP) based approaches and significantly improved QR code recognition success. Especially in scenarios with high motion blur, the recognition rate is reported to be 8–12% higher on average than traditional methods.

*Udvaros & Szabó (2024)* examined the role of image processing techniques in QR code recognition and decoding processes in detail. The study evaluates the effect of several preprocessing steps such as grayscale conversion, contrast enhancement, edge detection, morphological operations and perspective correction on QR code recognition success. In the experimental analysis of these methods, it was observed that the QR code recognition rate increased from 94% to 99% after correct segmentation and edge correction. The article demonstrates the success of these methods, especially in real-world conditions (low contrast, light reflection, *etc.*).

*Zheng et al. (2023)* proposed the Edge-Enhanced Hierarchical Feature Pyramid GAN (EHFP-GAN) model, a two-stage GAN-based structure for repairing damaged QR codes. The model preserves the integrity of the structure by reconstructing edge information and provides multi-scale context awareness with the hierarchical feature pyramid. Experiments demonstrated the superior performance of the model in terms of both visual quality and QR code recognizability. For example, the QR code recognition rate was reported to be 95.35% for mild deterioration, 79.84% for moderate deterioration and 31.94% for severe deterioration. These values outperformed the other GAN and classical restoration models compared.

*Ma, Chen & Wang (2023)* have analyzed the symbol ratio unique to QR codes prioritization and L2 norm minimization to develop a fast and blunt sharpening algorithm. In particular, the model focuses on preventing over-sharpening from negatively affecting the recognition process by distorting the structure of the symbols on the QR code. Furthermore, the processing time is reduced by proposing a recognizability metric based on "finder pattern" detection. In the experiments, the QR code recognition rate of the proposed method varied between 87% and 96% depending on the fuzziness level, and the results were 5% to 15% better than the traditional approaches. Moreover, the processing time is on average 30% shorter than conventional models.

In recent years, deep learning-based super-resolution models have emerged as an effective solution for enhancing the readability of low-quality QR codes. The EDSR model was designed to reconstruct fine details in low-resolution images through a deep residual network structure (*Zhang et al., 2022*). EDSR is particularly notable for its capability to restore high-frequency details in high-resolution images. Similarly, *Ko et al. (2021)* demonstrated that light field super-resolution methods could effectively recover fine details in QR codes through deep learning-based adaptive feature blending.

Another notable super-resolution model, VDSR, employs a deep network architecture with high computational capacity, enabling fast and accurate reconstruction of finely detailed images, including QR codes (*Wang et al., 2023*). The model's extensive layer depth provides an effective solution for addressing resolution-related challenges, particularly in QR codes with intricate details. Furthermore, ShuffleMixer, an efficient convolutional network, has shown significant success in enhancing low-resolution images, including QR codes, by optimizing the layer structure while maintaining a low computational cost (*Sun, Pan & Tang, 2022*).

The ESPCN model offers a fast and computationally efficient approach to enhancing the resolution of images containing fine details, such as QR codes, by utilizing sub-pixel convolutional layers (*Ko et al., 2021*). This model is particularly well suited for real time applications due to its ability to reduce data redundancy and computational overhead while preserving high resolution details. Meanwhile, SRCNN is a foundational model in super-resolution applications, improving QR code resolution and enhancing readability through convolutional layers (*Sun, Pan & Tang, 2022*).

*Ran et al. (2023)* introduced the GuidedNet model, which contributes significantly to the field of super-resolution. GuidedNet enhances the resolution of images containing fine details, such as QR codes, by leveraging high-resolution guidance to reconstruct lost information. This model holds strong potential for QR code applications, particularly in scenarios with high data density. Additionally, *Xiao et al. (2023)* developed the Local Global Temporal Difference Learning model for satellite video super-resolution, which improves resolution by considering both local and global variations. Given its capability in detail-oriented applications, this model may also be applicable to dynamic QR code processing (*Ran et al., 2023*).

*Edula et al. (2023)*, developed a method for detecting and reading fuzzy and low-resolution QR codes using YOLOv8 object detection and Real-ESRGAN super-resolution model. In the study, the YOLOv8 model was trained with five different

configurations, and QR codes were detected by selecting the model with the highest precision and recall. Then, the detected QR codes were optimized using Real-ESRGAN and tried to be read through Pyzbar, ZXing, and OpenCV libraries. The results of the study show that the proposed method offers a higher success rate compared to traditional QR code readers (*Edula et al., 2023*).

*Tanaka et al. (2024)*, developed a super-resolution method that emphasizes the importance of QR codes on the readability of the information in the module centers. Noting that traditional SRCNN and QRCNN methods are not optimized for QR codes, the researchers presented a training strategy specific to the modular structure of QR codes. The proposed model optimized the super-resolution process with a special focus on recovering the information in the centers of QR code modules. Experimental results show that the model trained on masked regions provides higher read rates compared to previous SRCNN and QRCNN methods (*Tanaka et al., 2024*).

*Shindo et al. (2022)*, proposed two deep learning-based models, QRCNN and QRGAN, to convert low-resolution QR codes to high-resolution. Noting that common super-resolution methods such as SRCNN, SRResNet and SRGAN are not specifically designed for QR codes, the researchers designed the QRCNN and QRGAN models to be lighter and faster. In experiments, it was shown that the QRGAN model achieved the highest QR code reading performance thanks to adversarial learning. The findings of the study reveal that customized super-resolution methods for QR codes can outperform general super-resolution models (*Shindo et al., 2022*).

*Kato et al. (2011)*, developed a multi-image super-resolution method to prevent QR codes from being unreadable due to low resolution. In this study, a special binary pattern constraint is added to the super-resolution process by considering the binary structure of QR codes. Compared to traditional super-resolution methods, this approach is shown to make QR code boundaries sharper and significantly improve the readability of the code. In particular, experiments on low-resolution QR codes show that the proposed method offers a 15.7% higher success rate compared to traditional super-resolution algorithms.

While existing work has explored super-resolution models to improve QR code resolution, most of it has focused on artificially generated low-resolution images rather than real-world distortions. This work addresses this gap by using a dataset derived from scanned images that capture distortions commonly encountered in practical applications, such as blur, contrast reduction and scanning artifacts. By systematically evaluating the performance of EDSR, VDSR, ESPCN and SRCNN under these realistic conditions, this research provides a more comprehensive assessment of super-resolution models in QR code restoration. The findings contribute to the development of more robust QR code enhancement techniques by improving readability in real-world scenarios where traditional QR code detection methods often fail. Furthermore, more challenging QR codes that even deep learning-based readers cannot read are simulated in a computerized environment and high improvements are achieved with robust models.

These contributions demonstrate the potential of super resolution technology for the reliable reading of low resolution and degraded QR codes and enable more effective use of QR codes in commercial and security applications.

## MATERIALS AND METHODS

In this study, the performances of four different super resolution models, EDSR, VDSR, ESPCN and SRCNN, are evaluated to improve the readability of low resolution QR codes. In this section, the dataset, super resolution models and performance evaluation metrics are described in detail.

### Dataset

The dataset of the study is divided into three main groups: real scanned data and computer simulated QR code data. This dataset consists of three folders. The first folder (Unreadable Scanned QR Codes) contains images obtained from the scanner and unreadable with OpenCV QRCodeDetector, the second folder (Learning Data) contains computer-generated QR code images created by simulating the scanner, and the third folder (Unreadable Simulated QR Codes) contains simulated QR Code images unreadable with OpenCV QRCodeDetector and WeChatQRCodeDetector. All images were created in JPG format. "Data.csv" file contains the data of the QR Codes. The dataset can be accessed on https://doi.org/10.6084/m9.figshare.28424213.v1.

#### Real scanned data

A total of 38,400 QR codes were printed on a printer and scanned at 300 dpi resolution. These scanned images, with 96 QR codes on each page, were cropped and saved in 224 × 224 size, and QR code reading was performed with OpenCV. Afterward, 4,593 QR codes were found to be unreadable. These unreadable QR codes were used as a test data set to evaluate the impact of super-resolution models on readability. Figure 1 shows images of the scanned QR codes that could not be read with Opencv.

#### Simulated QR codes data

The dataset contains 2,899 computer-simulated QR code images designed to test the models against significant distortions. These QR codes could not be read by either the OpenCV QRCodeDetector or the WeChat QRCodeDetector. Figure 2 illustrates examples of the simulated QR codes that were unreadable by both OpenCV detectors.

#### Train data

As training data, a total of 16,000 QR codes with high and low resolution were generated on computer. To increase precision, these QR codes, consisting of 12, 24, 36, and 48-character data, were generated at two different resolution levels to simulate the scanning process. Geometric operations such as rotation, zooming, and panning were applied for both classes so that the same transformations were performed in both classes. In the low-resolution class, blur, contrast changes, and noise were added to simulate

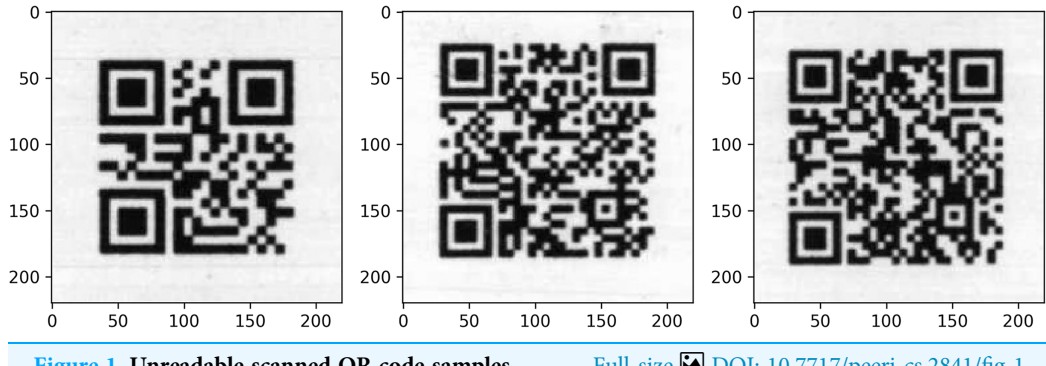

**Figure 1 Unreadable scanned QR code samples.**

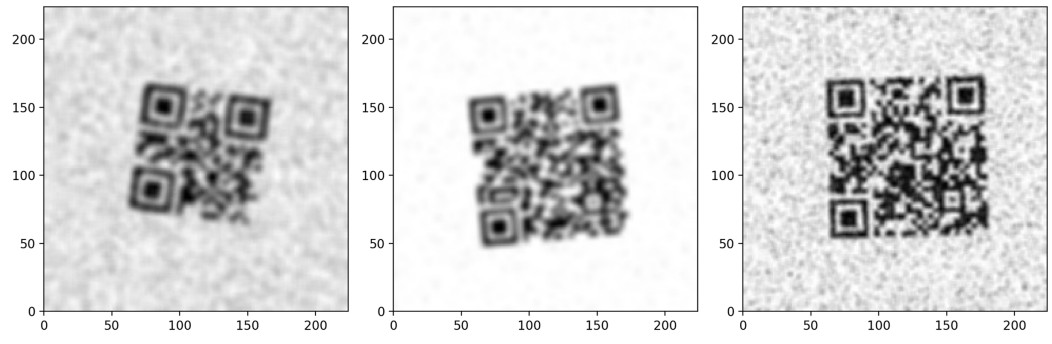

**Figure 2 Unreadable simulated QR code samples.**

scanner-induced distortions. This was done to ensure that the low-quality scans were representative of the distortions that would occur in real-world conditions. Figure 3 shows a visualization of the high and low-resolution classes of the dataset.

To simulate scanner distortions, we applied a series of augmentations to the low-resolution QR code images. The transformations included:

Affine transformations: Rotation (−10° to 10°), translation (−10 to 10 pixels in both x and y directions), and scaling (0.8 to 1.2). These were applied deterministically to introduce variations in positioning, orientation, and size. Additive Gaussian noise: Noise with a scale of up to 0.4 × 255 was added to mimic scanner-induced artifacts and distortions.

Linear contrast adjustment: The contrast was randomly modified within the range of 0.75 to 4 to simulate variations in scanner exposure and lighting conditions.

Gaussian blur: A blur effect with a sigma range of 1.5 to 3.0 was applied to degrade sharpness, imitating the loss of detail that occurs due to scanner quality and focus issues.

These augmentations ensured that the generated low-resolution QR codes closely resembled real-world scanned QR codes with various distortions, making the training process more robust. Figure 3 presents examples of simulated dataset samples used for training the model, which incorporate various degradations such as blur, noise, and contrast reductions to mimic real-world scanning conditions.

**Peer**J Computer Science

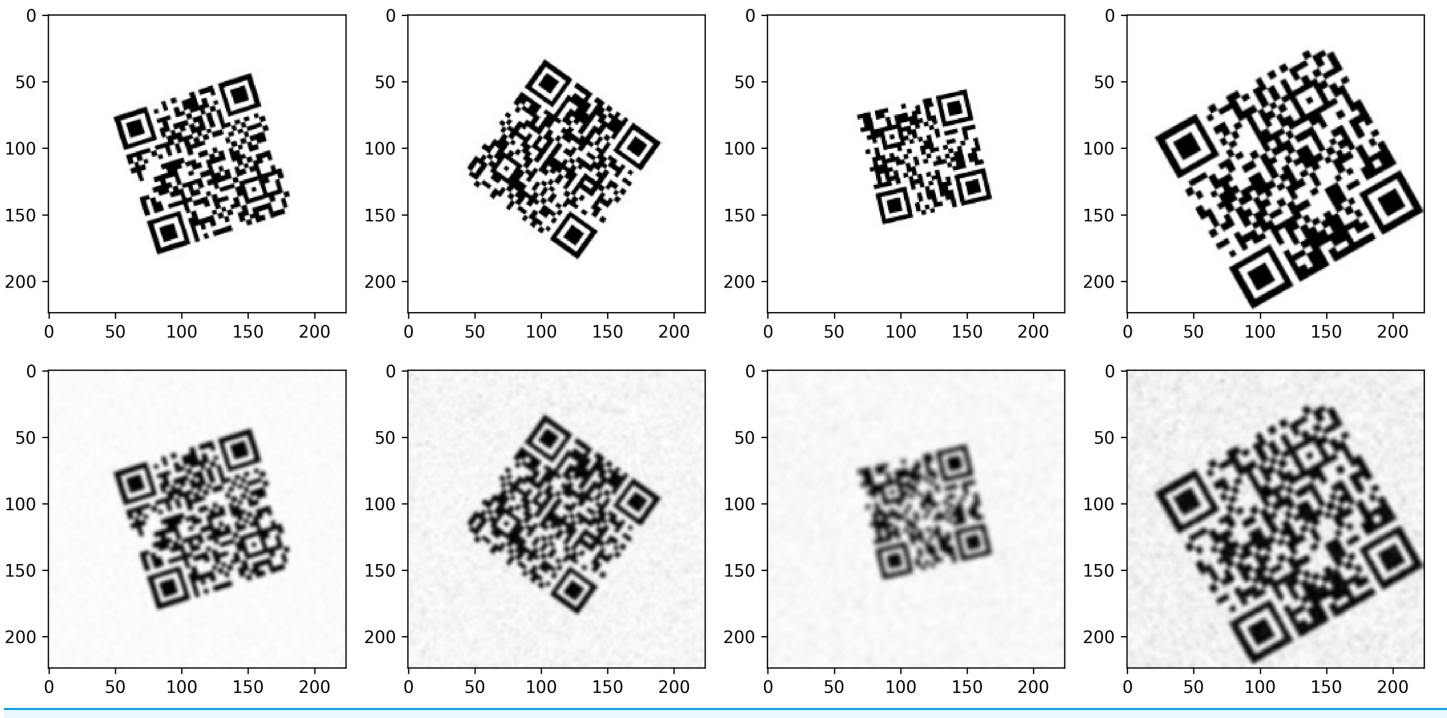

**Figure 3 Simulated dataset samples for train model.**

## Super resolution models

The four super-resolution models used in this study are as follows:

### *Enhanced deep super resolution*

Based on deep residual network structure, this model was developed to recover fine details in low-resolution images. EDSR aims to minimize resolution loss by improving the details of QR codes. In the VDSR model, each convolution layer processes the input sequentially.

The EDSR model incorporates residual blocks to improve feature learning and preserve fine details. The structure can be represented as:

- First convolution layer:

$$x_1 = \text{ReLU}(\text{Conv}(X, W_1) + b_1)$$

- Each residual block:

$$x_i = x_{i-1} + \text{BatchNorm}(\text{Conv}(\text{ReLU}(\text{Conv}(x_{i-1}, W_i)), W_{i+1}))$$

where $i$ denotes the residual block index.

- Final convolution layer with global residual connection:

$$Y = \text{Conv}(x_n, W_{\text{out}}) + x_1$$

where $x_1$ is the global residual connection from the first layer.

### Very deep super resolution

VDSR is a structure for achieving high quality in low resolution images with deep layers. Thanks to its high computational capacity, it performs successfully in detail intensive images such as QR codes.

In the VDSR model, each convolution layer processes the input sequentially. The overall structure of the model can be represented as follows:

- First convolution layer:

$$x_1 = \text{ReLU}(\text{Conv}(X, W_1) + b_1)$$

where $X$ is the input image, $W_1$ and $b_1$ are the weights and bias of the first convolution layer, respectively.

- Intermediate layers:

$$x_i = \text{ReLU}(\text{Conv}(x_{i-1}, W_i) + b_i)$$

for $i = 2, 3, \ldots, 18$.

- Final convolution layer:

$$Y = \text{Conv}(x_{18}, W_{19}) + b_{19}$$

where $Y$ represents the model's output.

### Efficient sub pixel convolutional neural network

ESPCN is suitable for real time applications by preserving data density with sub pixel convolution layers. It is especially preferred for increasing the readability of QR codes with its fast and efficient resolution increase capacity.

In the ESPCN model, the final layer performs a sub pixel convolution operation to upscale the input image and produce a high resolution output:

- First convolution layer:

$$x_1 = \text{ReLU}(\text{Conv}(X, W_1) + b_1)$$

- Second convolution layer:

$$x_2 = \text{ReLU}(\text{Conv}(x_1, W_2) + b_2)$$

- Final layer (sub-pixel convolution):

$$Y = \text{SubPixelConv}(\text{Conv}(x_2, W_3) + b_3)$$

Here, the sub pixel convolution rearranges the lower resolution feature map into a higher resolution output.

### Super resolution convolutional neural network

SRCNN is a basic model in the field of super resolution and quickly increases low resolution images to high resolution with its simple structure and efficient operation.

The SRCNN model has three convolution layers:

- First layer:

$$x_1 = \text{ReLU}(\text{Conv}(X, W_1, 9) + b_1)$$

where $X$ is the input, $W_1$ and $b_1$ are the weight and bias terms, and nine denotes the filter size.

- Second layer:

$$x_2 = \text{ReLU}(\text{Conv}(x_1, W_2, 5) + b_2)$$

- Third (final) layer:

$$Y = \text{Conv}(x_2, W_3, 5) + b_3$$

where $Y$ is the final output of the model.

## EXPERIMENTAL RESULTS

In this study, the performances of EDSR, VDSR, ESPCN and SRCNN models are compared to improve the readability of low resolution QR codes. The learning dynamics and generalization capacities of the models were analyzed by examining their losses in training and validation processes.

For training the models, EarlyStopping and ReduceLROnPlateau callback mechanisms were used to optimize the training process and avoid unnecessary epoch runs. EarlyStopping stopped training and restored the best weights if the validation loss metric did not improve with respect to its value for five patience periods. In this way, the risk of overfitting was reduced, and unnecessary computational costs were avoided. On the other hand, ReduceLROnPlateau was used to monitor the change in the verification loss and reduce the learning rate by a factor of one (factor = 0.1) if there was no improvement in the loss within two epochs. This approach allowed the model to be updated in smaller steps, making it easier to reach a better local minimum. The learning rate was initially initialized at 0.0001, and a lower bound of 0.000001 was set to prevent the learning rate from dropping too low. These mechanisms helped to make the training process of the model more efficient and improved the validation performance.

As shown in Fig. 4, the training and validation losses of the SRCNN model initially decrease rapidly, indicating that the model adapts quickly to the learning process. As the epoch progresses, both losses continue to decrease, but this process slows down gradually. The validation loss of the model is quite close to the training loss and does not show large fluctuations. This indicates that the model has a stable generalization capacity on the validation data. The EDSR model starts to learn quickly at the beginning of the training process, and a balanced reduction between training and validation losses is observed. The training and validation losses are quite close to each other, and as the epochs progress, the losses approach the minimum level. There are no serious fluctuations during the validation loss of the model, and it seems to have a stable learning process in general. The losses of the VDSR model decrease rapidly early in the training process, indicating that the model has a high capacity to learn the data. The training and validation losses are largely parallel, with small fluctuations in the validation loss as the epochs progress. The ESPCN model undergoes a rapid learning process and exhibits a rapid decrease in training and validation losses in the early epochs. From about epoch 16 onwards, the validation loss of the model stabilized and stabilized at very low levels. The training and validation losses are very close

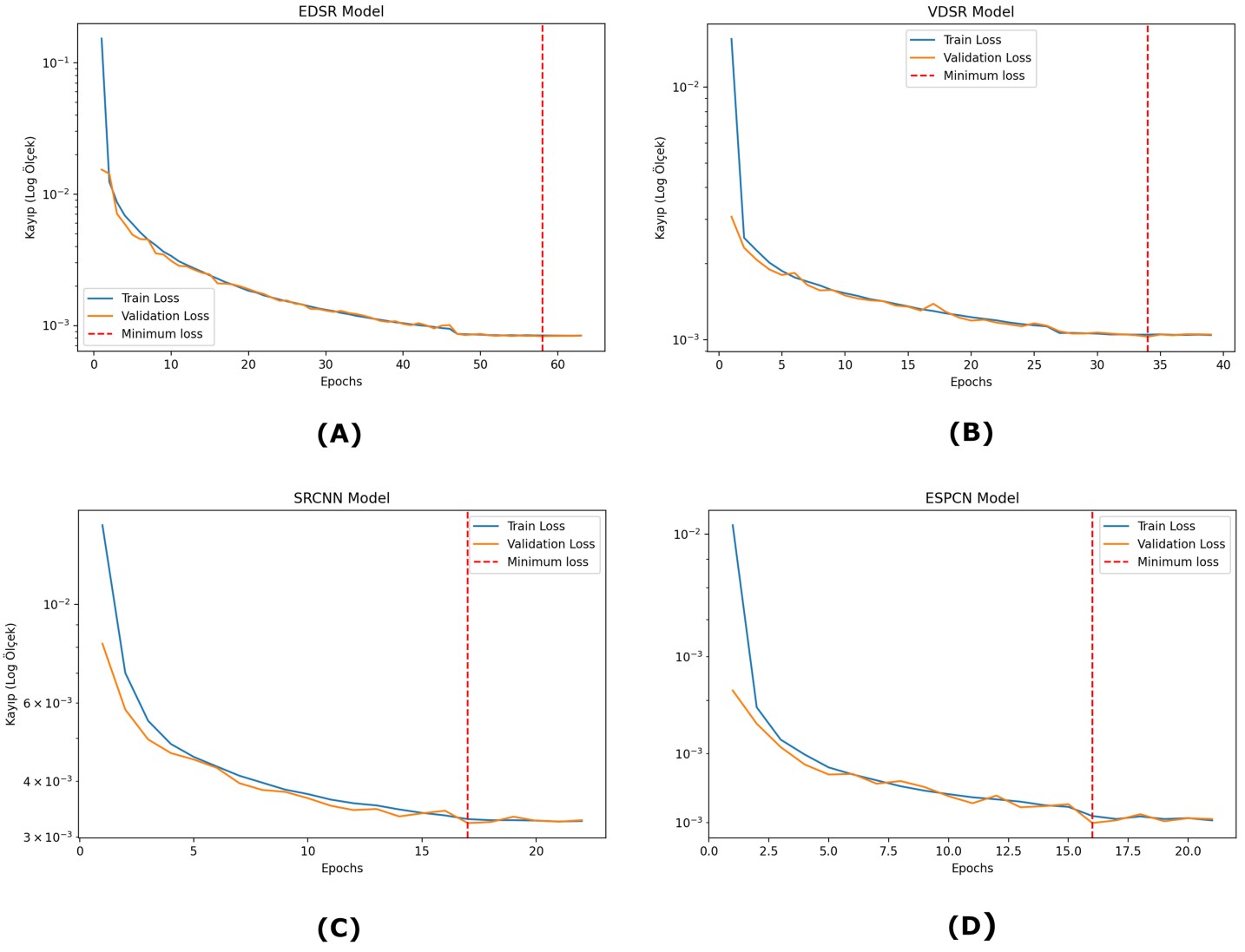

**Figure 4** **Training and validation loss graphs for each model.** (A) EDSR model (B) VDSR model (C) SRCNN model (D) ESPCN model.

to each other, and it is seen that the model has high generalization success with the validation data.

Figure 5 shows that the super-resolution applied to the scanned images with the respective models improves these results, each showing different performance characteristics. The original scanned image shows blurring and distortion, limiting the readability of the QR codes. The VDSR and EDSR models greatly improved the resolution of the QR codes, especially in sharpening the edge details and providing the best visual quality. On scanned QR codes, all four models significantly improved the quality.

Figure 6 shows the super-resolution results on a simulated blurred QR code image. The first image represents the degraded QR code, which exhibits significant blurring and

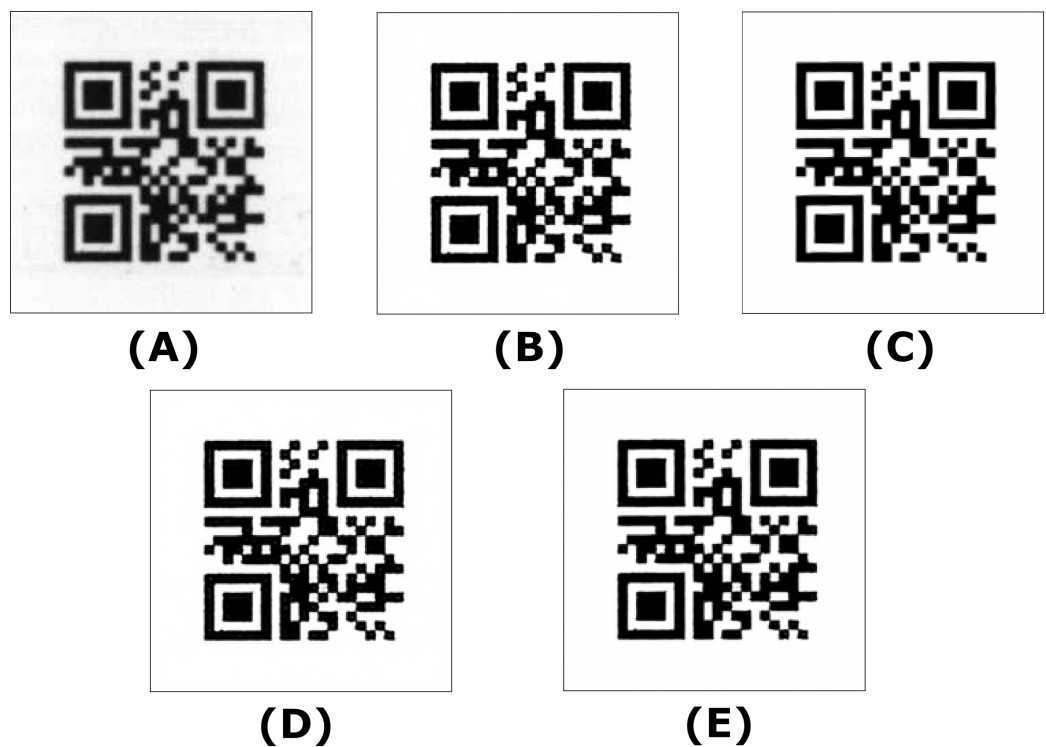

**Figure 5 Examples of trained models enhancing scanned QR code images.** (A) Sample scanned QR code. (B) Enhanced with EDSR model. (C) Enhanced with VDSR model. (D) Enhanced with SRCNN model. (E) Enhanced with ESPCN model. 

structural distortions and becomes unreadable. The next images show the reconstructions produced by four different super-resolution models. The EDSR model provided the highest level of sharpness, significantly improving the fine details of the QR code. Contrast is well preserved and the distinction between black and white regions is clear. This makes it one of the most effective models for QR code restoration. The ESPCN model offered a well-balanced reconstruction with minimal pixelation. The SRCNN model partially improved the QR code but struggled to recover finer details. Some areas remained blurred, which affected the successful decoding of the QR code. The VDSR model improved the image quality but caused minor distortions along the edges. Although it improved contrast and structure, some distortions that could affect readability could not be removed. For the simulated QR codes, the EDSR and VDSR models improved the quality relatively better when the distortion rate was very high.

Table 1 presents a comparative analysis of four different super-resolution models: VDSR, EDSR, ESPCN, and SRCNN, with respect to peak signal to noise ratio (PSNR), structural similarity index measure (SSIM), and model size. These metrics are crucial in evaluating the effectiveness of each model in improving the readability of degraded QR codes.

EDSR achieved the highest PSNR (35.23 dB) and SSIM (0.7509), indicating that it provides the most accurate reconstruction among the models. This shows that EDSR

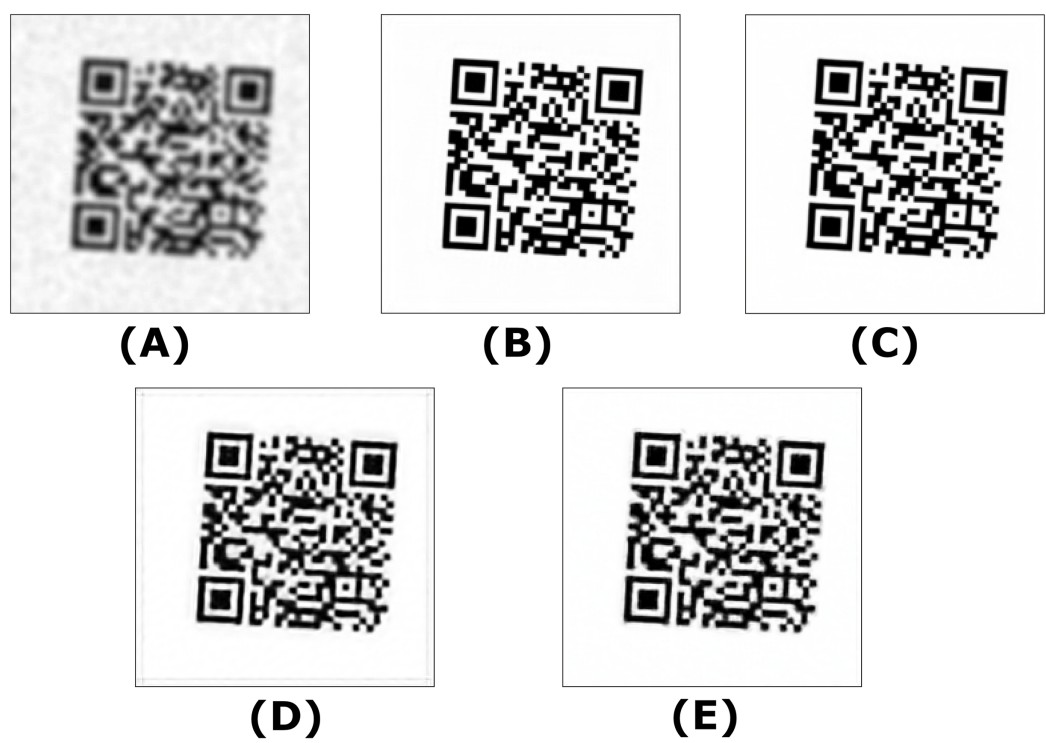

**Figure 6 Examples of trained models enhancing simulated QR code images.** (A) Sample simulated QR code. (B) Enhanced with EDSR model. (C) Enhanced with VDSR model. (D) Enhanced with SRCNN model. (E) Enhanced with ESPCN model.

**Table 1 Performance metrics of super resolution models.**

| Model | PSNR (dB) | SSIM | Model size (MB) |
|---|---|---|---|
| VDSR | 35.19 | 0.7506 | 8.07 |
| EDSR | 35.23 | 0.7509 | 16.9 |
| ESPCN | 34.74 | 0.7461 | 0.33 |
| SRCNN | 34.54 | 0.7437 | 0.78 |

preserves structural details and pixel intensity distributions better than other methods. However, the model size (16.9 MB) is significantly larger, making it computationally more expensive. VDSR performed comparably to EDSR with a PSNR of 35.19 dB and a SSIM of 0.7506. While providing similar reconstruction quality, it has a smaller model size (8.07 MB), making it a more efficient alternative. ESPCN showed slightly lower PSNR (34.74 dB) and SSIM (0.7461) compared to EDSR and VDSR. However, its compact model size (0.33 MB) shows that it is extremely lightweight, efficient, and suitable for real-time or embedded system applications. Although SRCNN provides the lowest performance (PSNR: 34.54 dB, SSIM: 0.7437), it still shows competitive results. The small model size (0.78 MB) makes it a suitable option when computational constraints are a priority.

**Table 2 QR code detection accuracy and processing time of super-resolution models. OpenCV Detector and WeChat Detector refer to the QR code detection algorithms used for reading the images.**

| Model | Real scanned data read rate (%) | | Simulated data read rate (%) | | Avg. processing time (ms) |
|---|---|---|---|---|---|
| | OpenCV detector | WeChat detector | OpenCV detector | WeChat detector | Super resolution + QRCode decode |
| VDSR | 92.07 | 100 | 90.92 | 99.48 | 219 |
| EDSR | 92.77 | 100 | 94.72 | 99.72 | 364 |
| ESPCN | 92.64 | 100 | 41.18 | 83.92 | 112 |
| SRCNN | 88.00 | 100 | 39.94 | 88.30 | 121 |

Overall, EDSR and VDSR offer the best QR code restoration quality, while ESPCN and SRCNN sacrifice some performance for significantly smaller model sizes. The results emphasize that there is a trade-off between reconstruction quality and computational efficiency and that the choice of model depends on specific application needs.

Table 2 presents both QR code detection accuracy and average processing time for different super-resolution models. The results show that while EDSR achieves the highest detection rate for simulated QR codes (94.72% with OpenCV and 99.72% with WeChat), it also has the longest processing time (0.364 s) and is therefore less suitable for real-time applications. In contrast, ESPCN and SRCNN provide significantly faster processing times (0.112 and 0.121 s, respectively), making them more efficient for real-time QR code development. However, their detection rates on simulated data are significantly lower than EDSR and VDSR. These findings suggest that there is an inverse relationship between accuracy and speed, where EDSR and VDSR are preferred for accuracy-oriented applications, while ESPCN and SRCNN are more suitable for time-sensitive real-time scenarios.

## DISCUSSION

This article evaluates the performance of various super-resolution models in enhancing low-quality QR code images, analyzing their strengths and weaknesses in terms of detail preservation, readability, computational cost and real-time performance. The results show that super-resolution models significantly improve the readability of QR code, especially under degraded conditions. However, the trade-off between improving image quality and computational efficiency should be carefully considered, especially for real-time applications.

The findings show that EDSR achieves the highest detection accuracy (94.72% with OpenCV Detector and 99.72% with WeChat Detector for simulated QR codes) and is the most effective model for preserving QR code details. However, the high computational cost (0.364 s per image) limits its applicability for real-time and mobile applications. VDSR achieves comparable accuracy (90.92% with OpenCV Detector and 99.48% with WeChat Detector for simulated QR codes) while offering slightly better computational efficiency (0.219 s per image), making it a balanced choice between quality and performance.

In contrast, ESPCN and SRCNN exhibit significantly lower processing times (0.112 and 0.121 s respectively), making them more suitable for real-time applications. In particular, ESPCN has the smallest model size (0.33 MB), positioning it as a viable option for low-power devices where efficiency is a priority. However, the detection rates for ESPCN and SRCNN on simulated QR codes are quite low (41.18% and 39.94%, respectively), indicating that although these models are computationally efficient, they can struggle with highly corrupted QR codes. However, on real scanned images, OpenCV Detector with EPSCN achieves 92.64% and SRCNN achieves 88%, providing a faster solution in scenarios with low distortion.

These findings are in line with previous research highlighting the trade-off between computational efficiency and detail preservation in super-resolution applications (*Shi et al., 2022*; *Ran et al., 2023*). While deep learning-based methods have been widely used for photographic image enhancement, their application to QR code readability remains a new area of study. The results of this study contribute to filling this gap by systematically analyzing the impact of different super-resolution models on QR codes and providing insights into their practical usability.

While previous work have primarily relied on interpolation-based or shallow learning techniques, this study leverages deep learning-based super-resolution models that are better suited to handling real-world distortions. In particular, EDSR and VDSR, which utilize deeper architectures with residual learning, demonstrate superior performance in restoring fine details of QR codes compared to traditional methods. Additionally, ESPCN and SRCNN, though computationally more efficient, provide a balance between enhancement quality and processing speed.

## CONCLUSION

This study evaluates the effectiveness of deep learning-based super-resolution models in enhancing the readability of low-quality QR code images, focusing on their impact on detection rates, computational efficiency, and real-time performance. The findings demonstrate that super-resolution techniques significantly improve QR code detection, particularly under degraded conditions. Among the tested models, EDSR achieved the highest detection accuracy (99.72% for simulated data) but had the longest processing time (0.364 s per image), making it less suitable for real-time applications. In contrast, ESPCN and SRCNN exhibited substantially lower processing times (0.112 and 0.121 s, respectively), positioning them as more viable solutions for real-time QR code recognition. However, their lower detection rates in simulated data highlight their limitations in handling severely degraded QR codes.

These results emphasize the trade-off between accuracy and computational efficiency in QR code enhancement. While EDSR and VDSR remain the most effective models for applications prioritizing high-quality reconstruction, ESPCN and SRCNN are preferable for real-time processing scenarios where computational speed is a critical factor.

Future research should further investigate the real-time applicability of super-resolution models in embedded systems, mobile devices, and large-scale deployments. Additionally, exploring hybrid approaches that dynamically select models based on real-time processing

constraints could lead to more adaptive solutions. Moreover, testing these models under diverse scanning conditions (*e.g.*, varying illumination, distortions, and noise levels) will be crucial for assessing their robustness in real-world environments. Expanding the dataset to include a wider range of QR code degradations could also enhance the generalizability of the findings.

By systematically evaluating the trade offs between quality and efficiency, this study provides a foundation for future advancements in QR code enhancement using super resolution models, contributing to both theoretical and practical aspects of deep learning based image reconstruction.

### Funding
The authors received no funding for this work.

### Competing Interests
The authors declare that they have no competing interests.

### Author Contributions
- Yasin Sancar conceived and designed the experiments, performed the experiments, analyzed the data, performed the computation work, prepared figures and/or tables, authored or reviewed drafts of the article, and approved the final draft.

### Data Availability
The data is available at figshare: sancar, yasin (2025). QR Code Dataset V2. figshare. Dataset. https://doi.org/10.6084/m9.figshare.28424213.v1.

### Supplemental Information
Supplemental information for this article can be found online at http://dx.doi.org/10.7717/peerj-cs.2841#supplemental-information.

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
