# Peer review of "Reconstructing unreadable QR codes: a deep learning based super resolution strategy"

_PeerJ Computer Science, doi:10.7717/peerj-cs.2841_

## Round 0.1 · original submission · Major Revisions

Be sure to carefully respond to all the comments from all 3 expert reviewers.

Reviewer 1 ·

Basic reporting

In order to solve the problem of decreased readability of QR codes when they are distorted, misaligned, translated, and rotated, the paper uses EDSR, VDSR, ESPCN, and SRCNN to perform super-resolution reconstruction of QR codes, thereby improving the recognizability of QR codes.

1.The QR code sample in Figure 1 can be directly read by common app software such as WeChat, why can't it be read by OpenCV?

2.The convergence curve of VDSR is very strange. Please explain it.

3.This paper focuses on the super-resolution reconstruction algorithm used to improve the clarity of QR code images. In this scenario, in addition to image quality indicators, the real-time performance of the recognition algorithm should also be considered.

4.The grammar and spelling of the paper should be carefully checked throughout the paper.

5. In addition to deep learning super-resolution reconstruction algorithms, some non-deep learning deblurring and super-resolution reconstruction algorithms should be considered.

Experimental design

1.This paper focuses on the super-resolution reconstruction algorithm used to improve the clarity of QR code images. In this scenario, in addition to image quality indicators, the real-time performance of the recognition algorithm should also be considered.

Validity of the findings

1.The convergence curve of VDSR is very strange. Please explain it.

2.This paper focuses on the super-resolution reconstruction algorithm used to improve the clarity of QR code images. In this scenario, in addition to image quality indicators, the real-time performance of the recognition algorithm should also be considered.

·

Basic reporting

Clear and Unambiguous Professional English: It would help to clarify what “small details” refer to in real-time applications (line 44) and to specify which filters (e.g., blur, noise) were used to simulate scanner distortions (lines 84-85).
Literature References and Field Background/Context: It would be beneficial to explain more explicitly how your study fills a gap in QR code restoration research (lines 56-58).
Raw Data Shared: Adding metadata or a description of the dataset format would help ensure replicability.
Self-contained with Relevant Results to Hypotheses: A clearer connection between results and research hypotheses would strengthen the paper.

Experimental design

Original Primary Research: It would be beneficial to explain how this study specifically fills the gap in QR code restoration (lines 49-50).
Rigorous Investigation: More details on dataset specifics (e.g., image resolution, distortions) would improve reproducibility (lines 131-134).

Validity of the findings

Impact and Novelty: Further discussion on how EDSR, VDSR, ESPCN, and SRCNN compare to prior models in QR code restoration would strengthen the novelty (lines 59-95).
Conclusions: Limit conclusions to findings directly supported by the results (lines 236-249).

Additional comments

This article provides a comprehensive and innovative approach to improving QR code readability using super-resolution models. The authors have effectively compared multiple models and presented well-supported results, contributing valuable insights to the field of image enhancement.

Reviewer 3 ·

Basic reporting

no comment

Experimental design

At present, the decoder in wechat can solve the problems raised in this paper, please make a comparison and analysis.

Validity of the findings

no comment

Additional comments

1. The image seeking pattern of QR code is used to resist the rotation decoding, please explain in detail how the method of hypervariable rate solves the rotational error.
2. Please analyze in detail the speed of decoding after adding the proposed model.

---

## Round 0.2 · Minor Revisions

Dear authors,
Thanks a lot for your efforts to improve the manuscript.
Nevertheless, some concerns are still remaining that need to be addressed.
Like before, you are advised to critically respond to the remaining comments point by point when preparing a new version of the manuscript and while preparing for the rebuttal letter.

Kind regards,
PCoelho

Reviewer 1 ·

Basic reporting

1. What does Detec- in Table 2 mean? Why is there no data?

2. There are not enough references on QR Code in recent years in the paper

Experimental design

Although some experiments have been added, it is recommended to add 1-2 more comparison methods

Validity of the findings

It is innovative

·

Basic reporting

The manuscript is well-written, with clear and professional language.
The background now effectively highlights the research gap.

Experimental design

The study is well-structured, with improved dataset details and methodology.

Validity of the findings

he discussion strengthens the comparison of super-resolution models.
Conclusions are well-supported by the results.

Reviewer 3 ·

Basic reporting

1. Please check the format of the paper, such as Figure 6. Should there be a horizontal line there?
The format of the references needs to be checked and please check the title in the cited reference. Such as qr, cnn, pso. Whether they should be capitalized.
Whether the display format of the picture should be unified, for example, the subgraph in Figure 4 is represented by (A), (B), (C), and (D), whereas Figure 1 and Figure 2 are not.
Please check lines 105,112,118,125. Is it “et al.” or “et al”.
Please check that the first abbreviation does not write the full name. Such as line 41.
2. It is recommended to introduce the motivation of this paper in the introduction.

Experimental design

In Figure 2, I use WeChat version 8.0.56, the first QR code and the second QR code cannot be decoded, however, the third QR code can be decoded. Please explain if the third QR code should be decoded correctly.

Validity of the findings

Please explain in detail the differences between subgraphs (B), (C), (D), and (E) in Figures 5 and 6 respectively.

Additional comments

NO

---

## Round 0.3 · Minor Revisions

Dear authors,
Once more, thanks a lot for your efforts to improve the manuscript; some concerns are still remaining that need to be addressed.

Like before, you are advised to critically respond to the remaining comments point by point when preparing a new version of the manuscript and while preparing for the rebuttal letter.

Kind regards,

PCoelho

Reviewer 1 ·

Basic reporting

The relevant references on QR code deblurring, restoration and recognition are incomplete, especially the relevant literature in the past two years.

Experimental design

OK

Validity of the findings

OK

Additional comments

The relevant references on QR code deblurring, restoration and recognition are incomplete, especially the relevant literature in the past two years.

Reviewer 3 ·

Basic reporting

No

Experimental design

No

Validity of the findings

No

Additional comments

No

---

## Round 0.4 · accepted · Accept

Dear authors, we are pleased to verify that you meet the reviewer's valuable feedback to improve your research.

Thank you for considering PeerJ Computer Science and submitting your work.

Kind regards
PCoelho